# Human cytomegalovirus IE1 protein alters the higher-order chromatin structure by targeting the acidic patch of the nucleosome

Qianglin Fang[1,2†], Ping Chen[1†], Mingzhu Wang[1], Junnan Fang[1,2], Na Yang[1,2], Guohong Li[1,2*], Rui-Ming Xu[1,2*]

[1]National Laboratory of Biomacromolecules, Institute of Biophysics, Chinese Academy of Sciences, Beijing, China; [2]College of Life Sciences, University of Chinese Academy of Sciences, Beijing, China

**Abstract** Human cytomegalovirus (hCMV) immediate early 1 (IE1) protein associates with condensed chromatin of the host cell during mitosis. We have determined the structure of the chromatin-tethering domain (CTD) of IE1 bound to the nucleosome core particle, and discovered that the specific interaction between IE1-CTD and the H2A-H2B acidic patch impairs the compaction of higher-order chromatin structure. Our results suggest that IE1 loosens up the folding of host chromatin during hCMV infections.

*For correspondence:
liguohong@sun5.ibp.ac.cn (GL);
rmxu@sun5.ibp.ac.cn (RMX)

†These authors contributed equally to this work

Competing interests: The authors declare that no competing interests exist.

## Introduction

In eukaryotes, nuclear DNA is highly packaged into chromatin by histones. The nucleosome, the basic repeating unit of chromatin, typically assembles 146–147 bp of DNA wrapped around a histone octamer consisting of two copies each of H3, H4, H2A and H2B (*Luger et al., 1997a*). Nucleosomes connected by linker DNA form a 10-nm array resembling "beads-on-a-string". The binding of linker histone further compacts the linear array into a more condensed 30-nm chromatin fiber. The interaction between the N-terminal tail of histone H4 and a specific surface region on the neighboring nucleosome termed the "acidic patch" plays crucial roles for the formation of 30-nm chromatin fiber (*Dorigo et al., 2004*; *Luger et al., 1997a*; *Song et al., 2014*). The acidic patch is formed by a number of negatively charged residues of H2A and H2B, including Glu56, Glu61, Glu64, Asp90, Glu91 and Glu92 of H2A, and Glu102 and Glu110 of H2B (*Luger et al., 1997a*). Several nucleosome-binding proteins have been shown to specifically interact with the acidic patch. Therefore, it has been hypothesized that these proteins may play a role in regulating the higher-order chromatin structure by competing with the H4 N-terminal tails for binding to the acidic patch of the nucleosome (*Kalashnikova et al., 2013*; *McGinty and Tan, 2015*).

The 72-kDa immediately early 1 (IE1) protein of human cytomegalovirus (hCMV) plays critical roles in the viral early gene expression and DNA replication (*Gawn and Greaves, 2002*; *Mocarski et al., 1996*). In addition, IE1 has been long known to associate with condensed host chromatin during mitosis (*Ahn et al., 1998*; *Dimitropoulou et al., 2010*; *Huh et al., 2008*; *Krauss et al., 2009*; *Lafemina et al., 1989*; *Nevels et al., 2004*; *Reinhardt et al., 2005*; *Shin et al., 2012*; *Wilkinson et al., 1998*). The chromatin-tethering domain (CTD) located at the very C-terminal end of IE1 (a.a 476–491) is responsible for the association with the mitotic chromosome, specifically through the acidic patch of the nucleosome (*Mucke et al., 2014*; *Reinhardt et al., 2005*). However, the molecular mechanism underlying the association and the impact on the structure and function of

**eLife digest** Most of the DNA in a cell is tightly wrapped around groups of proteins called histones, which gives the impression of beads on a string. These bead-like structures are called nucleosomes, and interactions between histones in different nucleosomes can link one nucleosome to another, to package the DNA into a very condensed form.

Viruses sometimes interact with this condensed DNA; for example, a virus called human cytomegalovirus is known to attach to condensed DNA when cells are preparing to divide. But the consequences of these interactions are not always clear.

Now, Fang, Chen et al. have worked out the three-dimensional structure of a protein from the cytomegalovirus while it is attached to a nucleosome. This structure revealed that the viral protein connects to same part of the histones that otherwise helps pull the nucleosomes together.

Further experiments then compared how the cytomegalovirus protein attaches to nucleosomes with the interaction between nucleosomes and a similar protein from a different virus. Both viral proteins were seen to interact with the same part of the histone protein, but in different ways.

Next, Fang, Chen et al. showed that the DNA is more loosely packed when the cytomegalovirus protein is attached to the nucleosomes. This was not the case for the similar protein from the other virus. The experiments show that small differences in the ways viral proteins interact with condensed DNA can change their effects on DNA packaging. Additionally, these findings may help scientists to better understand how the binding of the cytomegalovirus protein to the nucleosomes might affect this virus's ability to infect or cause illness in humans.

host chromatin by IE1 remain to be determined. Here we provide an analysis of the structural basis for the interaction between the CTD of IE1 (IE1-CTD) and the nucleosome core particle (NCP) and explore the impact of their binding on the higher-order structure of chromatin.

## Results and discussion

The complex of NCP with an IE1-CTD peptide (a.a. 476–491) was obtained by soaking the peptide into preformed NCP crystals, and a 2.8 Å structure was solved by molecular replacement. The structure shows one molecule of IE1-CTD bound to the NCP (*Figure 1A*; *Figure 1—figure supplement 1*). The presence of only one IE1-CTD peptide in the complex structure is due to the availability of only one side of the NCP surface in the preformed crystal lattice, as in the case of NCP in complex with the latency-associated antigen (LANA) of Kaposi's sarcoma-associated herpes virus (KSHV) (*Barbera et al., 2006*). The IE1-CTD peptide adopts an extended, v-shaped conformation with a short α-helix at its C-terminus. In the complex structure, IE1-CTD is well positioned into the acidic patch of the nucleosome formed by H2A and H2B (*Figure 1A and B*). IE1-CTD contacts histone H2B at two distinct sites, the C-terminal portion of α1 and the N-terminal half of αC, through van der Waals interaction and intermolecular hydrogen bonds via its mainchain groups. Specifically, the amide and carbonyl groups of Thr480 make hydrogen bonds with the mainchain carbonyl and the sidechain amide groups of Gln44 of H2B (amino acid residue numbering following that in the reference of *Luger et al., 1997a*); the amide and carbonyl groups of Val484 bond the carboxylate group of Glu110 and the nitrogen atom of the imidazole ring of His106, respectively (*Figure 1B*). IE1-CTD contacts histone H2A via a number of sidechain contacts. His481 makes one hydrogen bond with Glu56 of H2A; Thr485 and Ser487 each makes one hydrogen bond with Glu64 of H2A; and Arg486 makes hydrogen bonds with Glu61 on α2, as well as with Asp90 and Glu92 located on αC of histone H2A. In addition, Met483 of IE1-CTD is placed in a hydrophobic pocket consisting of Leu23, and the aliphatic portion of Glu56 and Tyr57.

The nucleosomal acidic patch is well known for hosting the binding of a number of proteins (*Armache et al., 2011*; *Arnaudo et al., 2013*; *Barbera et al., 2006*; *Kato et al., 2013*; *Makde et al., 2010*; *McGinty et al., 2014*; *Wang et al., 2013*; *Yang et al., 2013*). Most closely related to hCMV IE1 is LANA of KSHV (*Barbera et al., 2006*). The N-terminal CTD of LANA forms a hairpin-like structure that pokes into the acidic patch of the nucleosome (*Figure 1C*). The distinctly folded IE1-CTD and LANA-CTD share certain common features of nucleosome binding. Structural

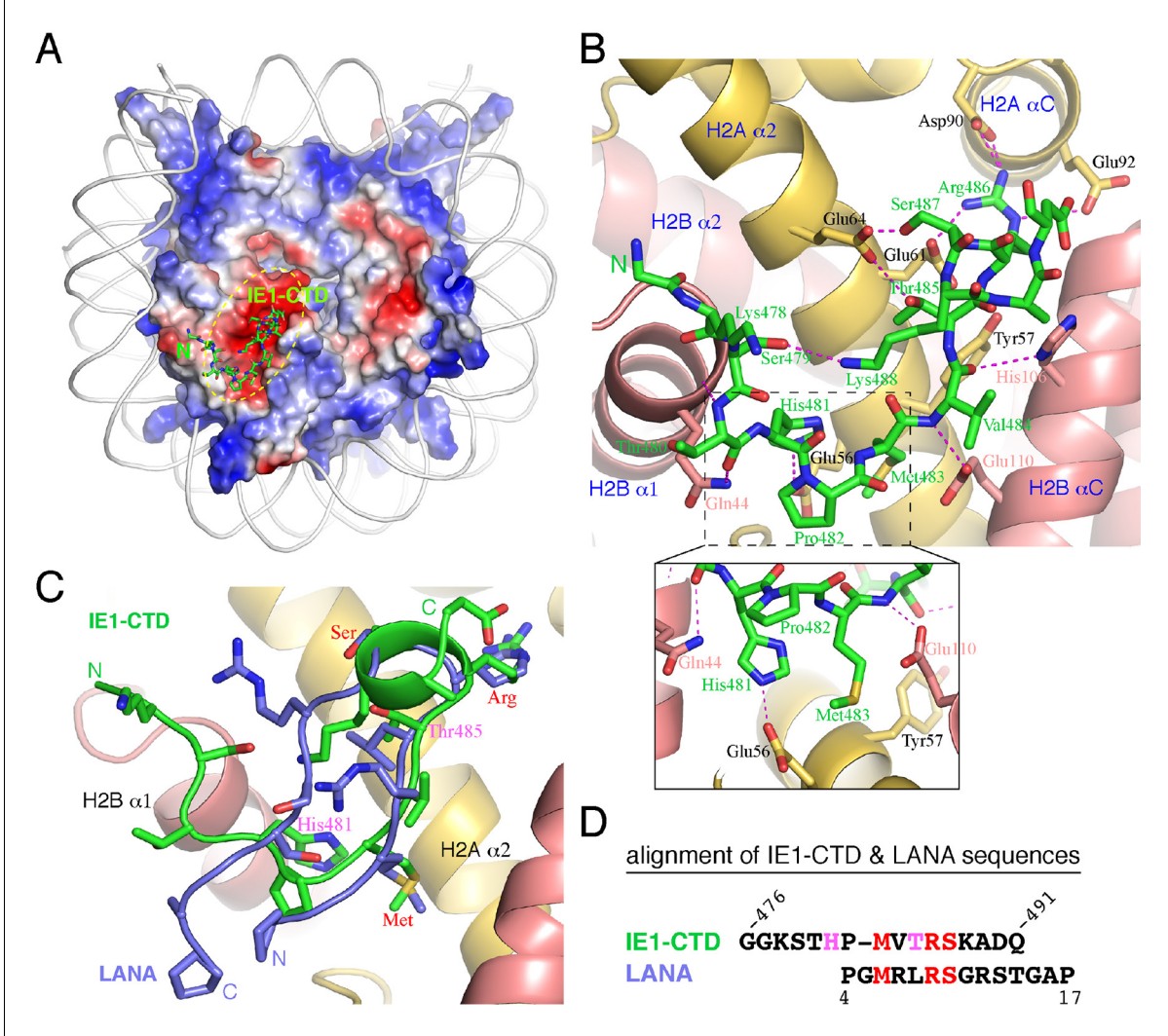

**Figure 1.** Structure of the IE1-CTD–NCP complex. (**A**) Location of the IE1-CTD binding site on the acidic patch of the nucleosomal surface. The histone octamer is shown as a surface representation colored according to electrostatic potential distribution (positive, blue; neutral, white; negative, red). DNA is shown as a cartoon colored white, and IE1-CTD is shown as a stick model. (**B**) A detailed view of the interaction between IE1-CTD and NCP. Histones H2A and H2B are shown in a ribbon representation superimposed with selected residues (in sticks) involved in interaction with IE1-CTD. Dashed lines indicate hydrogen bonds. An enlarged view of the region surrounding His481 of IE1-CTD is shown in an inset at the bottom of the figure. (**C**) Superposition of IE1-CTD and LANA. Both peptides are shown as a ribbon representation superimposed with sidechains (IE1-CTD, green; LANA, blue). (**D**) Structure-based alignment of IE1-CTD and LANA sequences. Residues colored in red are involved in similar interactions with the histones, and the two residues colored in magenta are engaged in IE1-CTD-specific interactions with histone H2A.

The following figure supplement is available for figure 1:

**Figure supplement 1.** An omit electron density map of the bound IE1-CTD.

comparison reveals that three IE1-CTD residues, Met483, Arg486 and Ser487, occupy the same regions of the acidic patch and engage the same sets of histone residues as the corresponding LANA residues in binding the nucleosome (*Figure 1C, D*, *2A and B*). In particular, Arg486 interacts with the acidic patch in a manner commonly found in the structures of protein-nucleosome complexes known to date (*Figure 2A–F*). A careful examination of the acidic patch reveals that it can be divided into three adjoining ligand-binding zones (*Figure 2C*). Zone I is formed by Glu61, Leu65, Asp90 and Glu92 of histone H2A, and Glu102 and Leu103 of histone H2B; Zone II is formed by histone H2A residues Tyr57, Ala60, Glu61 and Glu64, and the latter serves as a ridge separating zone I and II; and zone III is formed by Glu56 and Ala60 of H2A, and Val41, Gln44 and Glu110 of histone

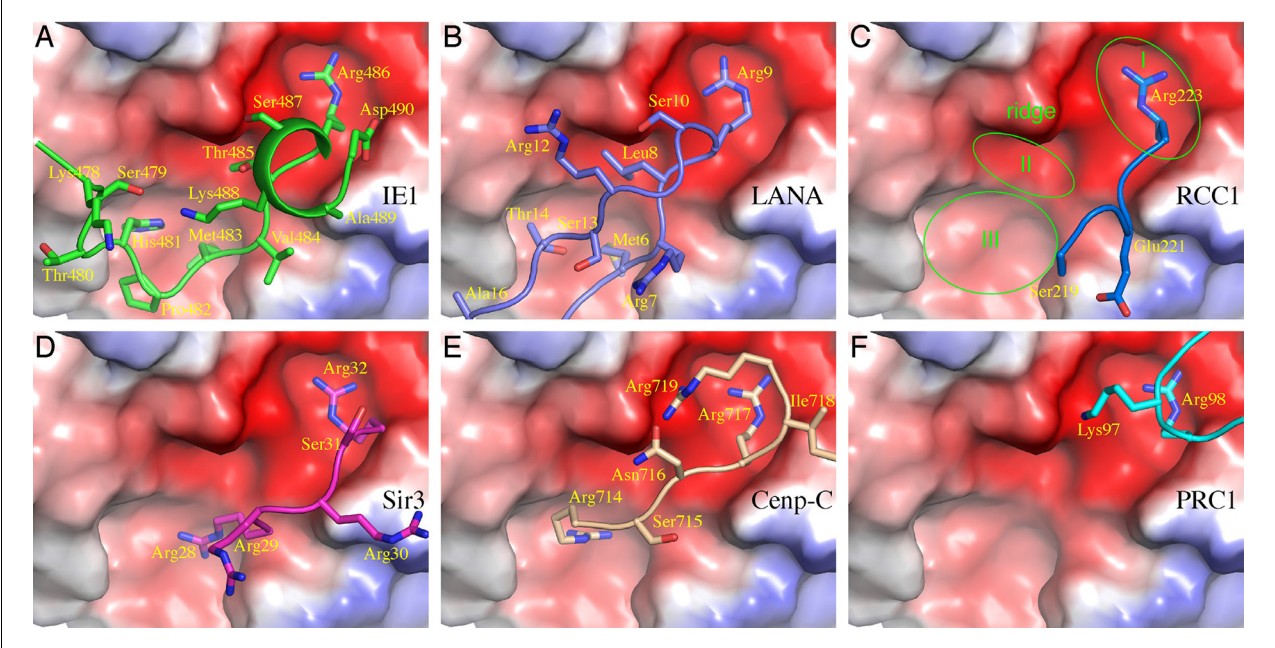

**Figure 2.** Comparison of protein binding modes to the acidic patch of NCP. All NCP-binding peptides or protein segments, shown in a stick model superimposed onto a cartoon representation of the backbone, were superimposed onto the structure of NCP, shown in a surface representation colored according to electrostatic potential, of the IE1-CTD complex based on alignment of NCP structures. (**A**) The binding of IE1-CTD to NCP. (**B**) LANA (PDB id: 1ZLA). (**C**) RCC1 segment (PDB id: 3MVD). The green ovals indicate distinct binding zones of the acidic patch. The protruding ridge at the junction between zone I and zone II is also labeled. (**D**) Sir3 (PDB1d: 4KUD). (**E**) CENP-C (PDB id: 4X23). (**F**) PRC1-RING1B (PDB id: 4R8P).

H2B. The binding of an arginine in zone I is conserved among all protein-NCP complexes known thus far. Thr485 of IE1 and Leu8 of LANA are bound in zone II, which is unoccupied in other NCP complexes (*Figure 2*). Main differences accounting for the specific interaction between IE1-CTD and NCP appear to reside in His481, which is bound in zone III and makes a hydrogen bond with Glu56 of H2A, and Thr485, which is bound to zone II and forms a hydrogen bond with Glu64 of H2A. And finally, the binding of the N-terminal segment of IE1-CTD spanning residues 476–480 to α1 of H2B is unique to IE1 (*Figure 1B*).

To reveal the determinants for the binding specificity of IE1-CTD, we carried out structure-guided mutagenesis of IE1-CTD and histones and analyzed their interactions using isothermal titration calorimetry (ITC). Wild-type IE1-CTD bound to recombinant human NCP with a dissociation constant ($K_D$) of 0.4 μM, while its mutant lacking the N-terminal segment interacting with α1 of H2B (Δ476–480) reduced the nucleosome-binding affinity to a $K_D$ of 12 μM (*Figure 3A and B*). To put the binding affinities in perspective, an approximately five-fold weaker binding of full-length IE1 than IE1-CTD alone to NCP was observed (*Figure 3—figure supplement 1A and B*), possibly caused by self-inhibitory effects of other domains in the full-length protein. In comparison, IE1 lacking the CTD showed no detectable binding to NCP (*Figure 3—figure supplement 1C*). With IE1-CTD, the most severe reduction of binding affinity was seen in the H481A mutant, which has a $K_D$ of 43.4 μM (*Figure 3C*), while loss of one hydrogen bond by substituting Thr485 with a valine brought the $K_D$ to 11.3 μM (*Figure 3D*). To gain further insights into the ~hundred-fold reduction of binding affinity introduced by the H481A mutation, we measured the effect of histone H2A mutation on Glu56, which interacts with His481 via hydrogen bonding and Met483 by hydrophobic/van der Waals interaction. ITC measurement shows that the NCP reconstituted with the E56R mutant of H2A, a charge-swap mutant, lowered the binding to IE1-CTD to a level beyond detection (*Figure 3E*). This change of histone H2A also brought about a conspicuous reduction of the binding of LANA to NCP from a $K_D$ of 0.25 to 23.4 μM (*Figure 3F and G*). LANA and IE1 share the methionine-mediated interaction with Glu56 in zone III of the acidic patch (*Figure 2A and B*), and the greater compromise of the binding of IE1-CTD to the H2A-E56R NCP reflects the importance of the His481-Glu56 hydrogen

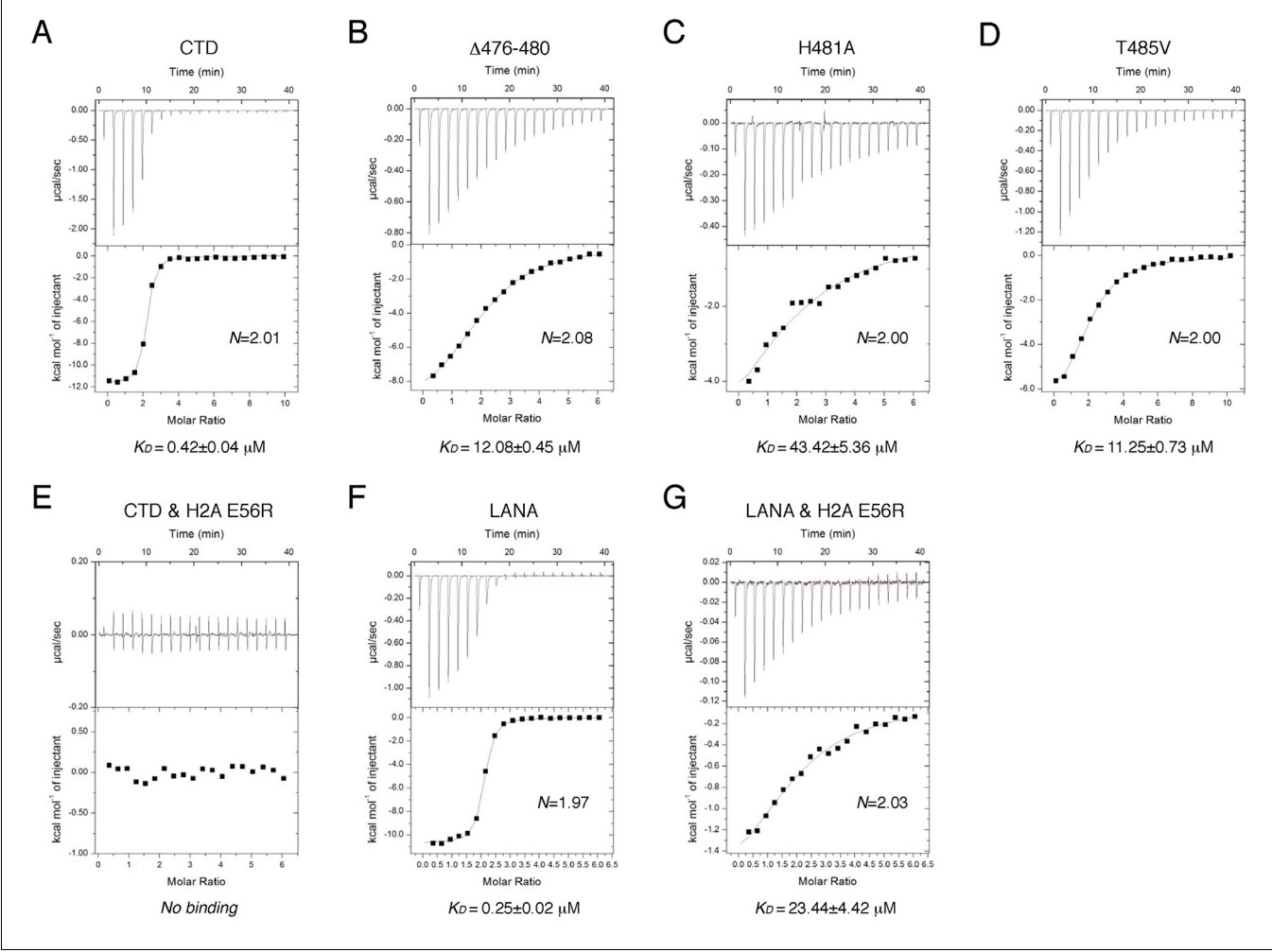

**Figure 3.** ITC measurements of peptide-NCP binding affinities. (**A–G**) Raw data and fitting curves of the integrated data for the indicated peptides and NCPs are shown together with the derived $K_D$ values and fitting errors.

The following figure supplement is available for figure 3:

**Figure supplement 1.** Binding of full-length IE1 to NCP.

bond in the IE1-NCP complex, as compared to the van der Waals interaction between Thr14 of LANA and Glu56 of histone H2A (*Figure 1B*).

The acidic patch of the nucleosome has been implicated in mediating higher-order chromatin folding via interaction with the N-terminal tail of histone H4 (*Luger et al., 1997a*; *Schalch et al., 2005*). Our previous cryo-EM structure of 30-nm chromatin fiber reveals that N-terminal tails of histone H4 are involved in inter-nucleosomal contacts between the tetranucleosomal structural units through the acidic patches of adjacent nucleosomes (*Song et al., 2014*). Since IE1-CTD is bound at the acidic patch, it is conceivable that IE1 binding may interfere with proper folding of the 30-nm chromatin fiber. To determine the extent by which the folding of chromatin fiber is affected by IE1 binding, we incubated IE1-CTD with the in vitro reconstituted 30-nm chromatin fiber assembled with an array of 12 tandem nucleosomes carrying repeats of 177 bp 601 DNA in the presence of linker histone H1, and analyzed the sample by analytical ultracentrifugation in sedimentation velocity (AUC) (*Song et al., 2014*). The nucleosomal arrays used for reconstituting 30-nm chromatin fiber were highly saturated and homogeneous, as examined by micrococcal nuclease (MNase) digestion and electron microscopy (*Figure 4—figure supplement 1A and B*). For comparison, the same batch of nucleosomal array (without H1) and 30-nm chromatin fiber (with H1) were used in AUC

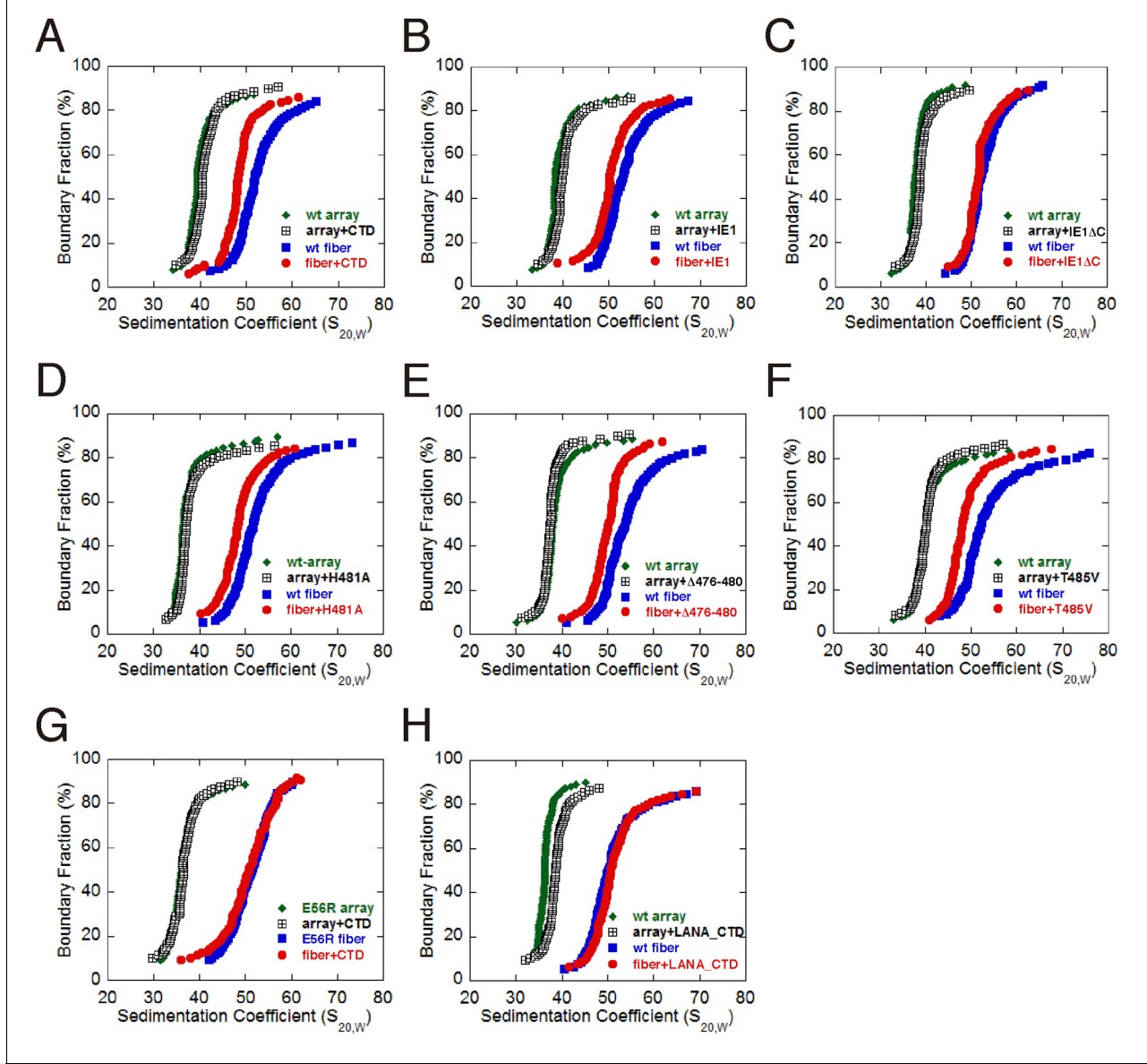

**Figure 4.** Influence of IE1-CTD on higher-order chromatin structure. (**A**) AUC analyses showing that IE1-CTD has little effect on the folding of the 10-nm nucleosomal array. Green and black data points represent that of 10-nm nucleosomal arrays in the absence and presence of IE1-CTD, respectively. In contrast, sedimentation profile of the 30-nm chromatin fiber reconstituted in the presence of linker histone H1 (blue squares) was shifted with the addition of IE1-CTD (red dots). (**B**) Full-length IE1 shares the property of IE1-CTD in selectively altering the folding of the 30-nm chromatin fiber. (**C**) A truncation variant of IE1 lacking CTD (IE1ΔC) does not alter chromatin structure. (**D–F**) Indicated IE1-CTD mutants retain the ability to impact the folding of the 30-nm chromatin fiber. (**G**) An E56R mutant of histone H2A renders IE1-CTD ineffective in altering the structure of the 30-nm chromatin fiber. (**H**) LANA-CTD (red dots) does not affect the folding of the 30-nm chromatin fiber. Instead, the 10-nm nucleosome array appears to be slightly affected with the addition of LANA-CTD peptide.

The following figure supplement is available for figure 4:

**Figure supplement 1.** Assessment of the quality of reconstituted nucleosomal array.

analysis. AUC experiments showed that, in the absence of IE1-CTD, the nucleosomal array sedimented with a median sedimentation coefficient $S_{ave}$ (sedimentation coefficient at 50% boundary fraction) of 36 ± 1 S, and the 30-nm chromatin fiber sedimented at 51.5 ± 0.6 S (*Figure 3A*). In the presence of IE1-CTD, the 10-nm nucleosomal array was unaffected while $S_{ave}$ of the 30-nm chromatin

fiber shifted from 51.5 to 48 S, indicating that the binding of IE1-CTD made the chromatin fiber more loosely folded (*Figure 4A*). It should be emphasized that the IE1-CTD-containing chromatin fiber represents an altered chromatin state different from both the extended 10-nm nucleosomal array and the folded 30-nm chromatin fiber. This chromatin-alteration property of IE1-CTD is shared by the full-length IE1 and fully depends on the presence of CTD (*Figure 4B and C*). Further AUC analyses with IE1-CTD mutants showed that they essentially retained the ability to decondense the 30-nm chromatin fiber, possibly due to their incomplete loss of NCP-binding abilities (*Figure 4D, E and F*). By contrast, chromatin fibers reconstituted with the E56R mutant of histone H2A, previously shown to be unable to interact with IE1-CTD, displayed no alteration of chromatin folding by IE1-CTD (*Figure 4G*). These observations indicate that the binding of IE1-CTD at the acidic patch of the nucleosome modulates the higher-order structure of chromatin. It should be pointed out that not all acidic patch-binding proteins affect chromatin folding, as LANA does not alter the folding of 30-nm chromatin fiber in our AUC analysis (*Figure 4H*).

Our structural analysis revealed that IE1-CTD binds the acidic batch of NCP in a distinct manner. A careful analysis of the landscape of the acidic patch reveals that it can be divided into distinct binding zones that host specific amino acid residues. The conserved binding mode of an arginine in zone I of NCP is important for the association of all NCP-binding partners known to date, while other zones of the acidic patch are involved in differential binding of individual partners. For example, zone III of the acidic patch hosts the binding of His481 of IE1-CTD and Thr14 of LANA, respectively. These two residues interact with Glu56 of histone H2A differently in the structure: while the τ nitrogen of the imidazole ring of His481 makes a hydrogen bond with the carboxylate group of Glu56 of histone H2A at a distance of approximately 2.9 Å, Thr14 of LANA contacts Glu56 of H2A approximately 5 Å away via van der Waals interaction. This difference in peptide-NCP interactions perhaps account for our observations that an E56R mutant affected the binding of IE1-CTD more severely than that with LANA.

For the purpose of dissecting the functions of individual NCP binding partners, one would ideally like to be able to isolate histone mutants that affect the binding of one protein but not others. The most promising such sites appear to lie in zone II and III, as they are less frequently occupied among known NCP binding proteins (*Figure 2*). However, the task is more challenging to distinguish IE1 and LANA bindings, as they both interact with all binding zones of the acidic patches. Nevertheless, the identification of the E56R variant of histone H2A as an IE1-noninteracting mutant has already served as a useful tool for assessing IE1's ability to modulate the higher-order structure of chromatin. Interestingly, this chromatin-modulating activity is not shared by LANA. It was shown previously that the LANA peptide promotes the compaction of nucleosomal arrays, judged by an assay in which the folding of nucleosomal arrays was induced by magnesium ions (*Chodaparambil et al., 2007*). In qualitative agreement with the previous observation, we saw that the addition of LANA made the nucleosomal arrays sedimented slower. Nevertheless, unlike IE1-CTD, LANA has no effect on the preformed H1-containing 30-nm chromatin fiber. There are two possible reasons for the different behavior of LANA, one possibility is that LANA is bound to the chromatin fiber but did not cause any changes to the folding, and the other possibility is that LANA failed to bind 30-nm chromatin fiber. In either case, IE1 and LANA differ in their ability to affect the folding of 30-nm chromatin fiber. The contrasting properties of IE1 and LANA suggest that distinct modes of protein binding to the acidic patch of the nucleosome could exert differential influences to chromatin folding. The discovery of the chromatin modulating activity of IE1 should facilitate further mechanistic understanding of its functions in viral pathogenesis, particularly its impact on the chromatin structure of the host genome, such as transcriptional regulation.

## Materials and methods

### NCP preparation

To express histones, cDNA fragments encoding *Xenopus laevis* histones H3.3C and H4, H2A type-1 and H2B 1.1, human H3.1 and H4, human H2A type 1-B/E and H2B type 1-J were cloned into pCDFDuet-1 vectors (Novagen) to generate four bicistronic plasmids for co-expression of *Xenopus* and human H3-H4 and H2A-H2B pairs, respectively, in the BL21(DE3)-RIL strain of *E. coli* at 37°C. Bacterial cells overexpressing H3-H4 and H2A-H2B were mixed and lysed together. *Xenopus* and

human histone octamers were then purified as described (*Kingston et al., 2011*). Xenopus NCP was assembled with a 146-bp palindromic DNA fragment derived from human α-satellite DNA according to a described procedure (*Luger et al., 1997b*). Human NCP was assembled with the 147-bp Widom 601 DNA sequence and the human histone octamer following the same procedure. Human NCP carrying the E56R mutation of histone H2A was generated by mutagenesis using the TaKaRa MutanBEST kit (TaKaRa, China).

## Crystallization, data collection, structure determination and refinement

NCP reconstituted with *Xenopus* histones was crystallized by sitting-drop vapor diffusion at 16°C in a condition containing 50 mM sodium cacodylate, pH 6.2, 100 mM magnesium acetate, and 11% 2-methyl-2,4-pentanediol. The co-crystal structure was obtained from a NCP crystal soaked with a chemically synthesized IE1-CTD peptide (a.a. 476–491, SciLight Biotechnology) at 2 mg/ml for 24 hr in a buffer containing 50 mM sodium cacodylate, pH 6.4, 100 mM magnesium acetate, 24% 2-methyl-2,4-pentanediol and 5% trehalose.

X-ray diffraction data were collected at 100K at Beamline BL18U of Shanghai Synchrotron Radiation Facility (SSRF) using a Pilatus 6 M detector at a wavelength of 1.0308 Å, and the data was processed using the HKL2000 package (*Otwinowski and Minor, 1997*). The structure was solved by molecular replacement with PHASER (*McCoy et al., 2007*) using the *Xenopus* NCP structure (PDB ID: 1AOI) as the search model. The electron density for the IE1-CTD peptide was clear after refinement with REFMAC (*Murshudov et al., 1997*) allowing unambiguous building of the IE1-CTD model using COOT (*Emsley and Cowtan, 2004*). The model was then refined with PHENIX (*Adams et al., 2010* COOT). The $R_{work}$ and $R_{free}$ of the final model were 19.5% and 24.4%, respectively. Detailed statistics for crystallographic analyses are shown in *Table 1*.

## IE1 protein preparation

Plasmids for expressing the full-length hCMV (Towne) IE1 and its truncation variant lacking CTD (IE1ΔCTD, a.a. 1–475) were obtained from Dr. Michael Nevels (*Mucke et al., 2014*). Both fragments were expressed as a GST-fusion protein at 16°C in the BL21(DE3)-RIL strain of *E. coli*. They were purified with glutathione-Sepharose resins, followed by cleavage of the GST-tag and further purification with a HiTrap Q HP column (GE Healthcare).

## ITC measurement

ITC experiments with IE1-CTD and LANA peptides were performed at 20°C, with the peptide solutions titrated into human NCP solutions in a buffer containing 10 mM Tris-HCl, pH 7.5, and 50 mM NaCl. An NCP concentration of 0.02 mM was used in all experiments, except for the titration of wild-type IE1-CTD into wild-type NCP, in which case 0.018 mM NCP was used. The

**Table 1.** Statistics of crystallographic analysis.

| Data collection statistics | |
| --- | --- |
| wavelength (Å) | 1.0308 |
| space group | $P2_12_12_1$ |
| unit cell (Å) | a = 106.70, b = 109.47, c = 181.98 |
| resolution (Å) | 30.00–2.80 (2.90–2.80) |
| $R_{merge}$ | 0.133 (0.611) |
| I/σI | 12.5 (3.3) |
| Completeness (%) | 99.9 (100.0) |
| Total/Unique reflections | 346679/52496 |
| **Refinement statistics** | |
| $R_{work}$/$R_{free}$ | 0.195/0.244 |
| rmsd bonds (Å) | 0.008 |
| rmsd angles (°) | 0.935 |
| No. of Atoms | |
| Protein | 6116 |
| DNA | 5982 |
| Peptide | 104 |
| Ion | 4 |
| Water | 230 |
| B factor (Å²) | |
| Protein | 35.9 |
| DNA | 87.8 |
| Peptide | 56.9 |
| Ion | 47.5 |
| Water | 36.0 |
| Ramachandran plot | |
| favored | 750 (98.7%) |
| allowed | 8 (1.1%) |
| outlier | 2 (0.3%) |

peptide concentrations used were, wild-type IE1-CTD at 0.87 mM, T485V at 0.99 mM, and the rest of IE1-CTD mutants and LANA all at 0.59 mM. Detailed procedures follow a protocol published previously (*Yang et al., 2013*). For the set of ITC experiments involving full-length IE1 and IE1ΔCTD, a buffer containing 10 mM Hepes, pH 7.4, and 150 mM NaCl was used to minimize background heat generation. Background heat measured through titrating samples from the syringe into the buffer without NCP was subtracted from the integrated data. For comparison, the binding of IE1-CTD to NCP under the same condition was also measured. In the set of experiments, an NCP concentration of 0.015 mM was used, and the concentrations of IE1, IE1ΔCTD and IE1-CTD used were 0.51, 0.52 and 0.59 mM, respectively.

## Chromatin reconstitution and analyses

Recombinant human core histones were prepared as described above. Linker histone H1.4 and DNA templates of 12 tandem 177 bp repeats of the 601 sequence were cloned and purified, and reconstituted into chromatin as previously described (*Chen et al., 2013*; *Dyer et al., 2003*; *Li et al., 2010*; *Song et al., 2014*).

The reconstituted chromatin samples were subject to AUC analysis in a buffer containing 10 mM HEPES, pH 8.0 and 0.1 mM EDTA. All AUC experiments were performed with nucleosomal array and chromatin fiber concentrations at 0.25 µM, and peptides/proteins to NCP at 5:1 molar ratio, whenever applicable, on a Beckman Coulter ProteomeLab XL-I, and the data were analyzed using enhanced van Holde-Weischet analysis and the Ultrascan II 9.9 revision 1504 as previously described (*Chen et al., 2013*).

MNase digestion of nucleosomal arrays follows a procedure described previously (*Li et al., 2010*). In brief, chromatin sample containing the equivalent of 1 µg DNA were incubated with micrococcal nuclease (Sigma) in a 50 µl reaction (10 mM HEPES, pH 7.5, 25 mM KCl, 0.2 mM EDTA, 10% Glycerol and 2 mM $CaCl_2$) at 37°C for 4 min. The digestion was terminated by addition of 50 µl stop buffer (200 mM NaCl, 2% SDS, 10 mM EDTA). Each sample were treated with 0.1 mg/ml Proteinase K and reacted at 55°C for 45 min. MNase digested DNA was purified and analyzed on a 1.3% agarose gel.

EM analysis by metal shadowing with tungsten was performed as previously described (*Chen et al., 2013*). The samples were examined using a FEI Tecnai G2 Spirit 120 kV transmission electron microscope.

## Accession codes

The coordinates and diffraction data have been deposited in PDB under the accession code 5E5A.

# Acknowledgements

We thank Yuanyuan Chen and Zhenwei Yang for technical support in ITC experiments, staff scientists at beamline BL18U of SSRF for assistance with X-ray data collection, and Dr. Michael Nevels for the generous gift of IE1 expression plasmids. This work was supported by grants from the Ministry of Science and Technology of China (Grant 2015CB856200), Natural Science Foundation of China (Grants 31430018, 31521002, 91219202, 31471218 and 31210103914), and the Strategic Priority Research Program (Grant XDB08010100) of Chinese Academy of Sciences (CAS). PC and NY are also supported by the Youth Innovation Promotion Association of CAS.

# Additional information

## Funding

| Funder | Grant reference number | Author |
| --- | --- | --- |
| Ministry of Science and Technology | 2015CB856200 | Na Yang<br>Guohong Li |
| National Natural Science Foundation of China | 31430018, 31521002, 91219202, 31471218 and 31210103914 | Ping Chen<br>Na Yang<br>Guohong Li<br>Rui-Ming Xu |

Chinese Academy of Sciences XDB08010100 Na Yang

The funders had no role in study design, data collection and interpretation, or the decision to submit the work for publication.

### Author contributions

QF, Prepared nucleosomes and proteins, Performed crystallization and ITC measurements, Acquisition of data, Analysis and interpretation of data, Drafting or revising the article; PC, Performed AUC experiments and analyzed data, Acquisition of data, Analysis and interpretation of data, Drafting or revising the article; MW, Collected crystal diffraction data and solved the structure, Edited the manuscript, Acquisition of data, Analysis and interpretation of data; JF, Performed AUC experiments and analyzed data, edited the manuscript, Acquisition of data, Analysis and interpretation of data; NY, Edited the manuscript, Conception and design, Analysis and interpretation of data; GL, R-MX, Conception and design, Analysis and interpretation of data, Drafting or revising the article

### Author ORCIDs

Rui-Ming Xu, http://orcid.org/0000-0002-6537-4623

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
