## [Decision Letter]

Thank you for submitting your work entitled "Human cytomegalovirus IE1 alters the higher-order chromatin structure by targeting the acidic patch of the nucleosome" for consideration by *eLife*. Your article has been reviewed by three peer reviewers, one of whom is a member of our Board of Reviewing Editors. The evaluation has been overseen by the Reviewing Editor and Randy Schekman as the Senior Editor.

The reviewers have discussed the reviews with one another and the Reviewing editor has drafted this decision to help you prepare a revised submission.

Summary:

Fang et al. present a crystal structure of human cytomegalovirus (hCMV) immediate early 1 (IE1) protein chromatin-tethering domain (CTD) bound to the "acidic patch" of the *X. laevis* nucleosome core particle (NCP). The authors report sub-micromolar affinity of the IE1-CTD peptide to the *H. sapiens* nucleosome core particle and identify mutations that disrupt binding of the peptide to the NCP. Fang et al. discover that full-length IE1 as well as IE1-CTD can modulate the structure of a reconstituted 30-nm chromatin fiber. The authors present a structure of a hCMV IE1-CTD-NCP complex at 2.8 Å resolution. The crystallography is well done, with good refinement parameters; the peptide is well defined.

Through structural and biochemical comparison of the N-terminal CTD of LANA to IE1-CTD they expand our understanding of chromatin fiber modulation by acidic patch binders; in contrast to LANA-CTD, IE1 shows chromatin-modulating activity. This indicates that different acidic patch binding proteins could exert effects on the chromatin structure in distinct ways. Fang et al. provide structural and biochemical data for the role of IE1 in 30-nm chromatin fiber modulation and present their findings in a concise and comprehensible manner.

This is only the second structure of a viral protein bound to the nucleosome. Previously reported structure (Barbera et al., 2006) characterized latency-associated antigen (LANA) of Kaposi sarcoma-associated virus binding to the same region on the nucleosome. Fang et al., show that three residues that are identical between IE1 and LANA use the same mode of interaction with the acidic patch. Additionally IE1 makes several specific interactions. Because there is only a very limited number of NCP complex structures, and because the presented structure reveals new insights, publication in *ELife* will be considered after the following concerns have been addressed and some additional experimental observations are included that should strengthen the conclusions. The manuscript would also greatly benefit from extended Discussion.

Essential revisions:

1) IE1-CTD binds to the NCP at the acidic patch like all other crystallized chromatin factor-NCP complexes, but its conformation in the acidic patch shows little similarity to the already published structures. However, the authors should analyze their structural data beyond a comparison with the LANA peptide. The data interpretation should be expanded to other chromatin-binding factors that have been crystallized at the acidic patch (RCC1, PRC1, Sir3, HGMN2, CENP-C), in terms of structure and sequence conservation. The authors recognize that different acidic patch-binding proteins modulate the chromatin structure differently but do not provide an explanation for this. Please expand the Discussion significantly and be accurate in referring to published literature.

2) To further elucidate the specific interaction of IE1-CTD and NCP, the authors introduced the mutation H2A E56R to disrupt binding of IE1-CTD to the acidic patch. The ITC measurements showed no binding of the IE1 CTD to NCPs with this mutation. Nevertheless, binding of LANA to the acidic patch was also significantly disrupted. It would be better if the authors would use/identify a mutation in the NCP that exclusively disrupts the binding of IE1-CTD. Disrupting interaction with ɑ1 of H2B seems to be an obvious mutation target. Did the authors try this? If not we recommend this to be done as the results may strengthen the conclusion that the mode of binding is unique.

3) Fang et al. report the purification of the full-length hCMV IE1 protein and use the full-length protein in their analytical ultracentrifugation experiment (Figure 3). Unfortunately, they do not include the full-length IE1 in their ITC measurements. Inclusion of full-length IE1 in the ITC measurements would facilitate comparison of affinities with their respective effects in 30-nm chromatin fiber modulation. This should be done.

4) Several issues about the analytical ultracentrifugation experiments have to be addressed. The authors do not provide an explanation of what the observed difference (between 51.5 S and 48 S) means. Isn't 48 S representative of a condensed chromatin as well? 12_177_601 array in presence of H1 sedimented at 46.6 S in their 2014 Science paper. Where does the discrepancy come from? The statement in paragraph four, Results and Discussion, is not accurate. The mutants only have some moderate 'effect'. At what concentration was this experiment done? Above or below the Kd? As it stands, the effects of the peptide on higher order structure may be over-interpreted, or not significant at all. Further, the authors have to include careful quality control for their 12mer array to check for saturation – e.g., by EcoRV digestion. This is especially important for arrays with H1. These issues with the AUC experiments must be clarified before publication can be considered.

---

## [Author Response]

*1) IE1-CTD binds to the NCP at the acidic patch like all other crystallized chromatin factor-NCP complexes, but its conformation in the acidic patch shows little similarity to the already published structures. However, the authors should analyze their structural data beyond a comparison with the LANA peptide. The data interpretation should be expanded to other chromatin-binding factors that have been crystallized at the acidic patch (RCC1, PRC1, Sir3, HGMN2, CENP-C), in terms of structure and sequence conservation. The authors recognize that different acidic patch-binding proteins modulate the chromatin structure differently but do not provide an explanation for this. Please expand the Discussion significantly and be accurate in referring to published literature.*

We have revised the manuscript according to the reviewer’s advice. Specifically, we have carefully compared the structures of proteins bound to the acidic patch of NCP. The result is shown in a new figure, Figure 2, in the revised manuscript. We have compared the binding modes of all the proteins mentioned except HGMN2, as there is no co-crystal structure available and the model of its binding to NCP is not publically available. From this comparison, we distinguished different ligand binding zones of the acidic patch (Figure 2), and this identification helped the grouping of different sites of contact.

We refrained from an extensive Discussion in the original manuscript due to the consideration of strictly following the journal’s length limitation. Following the reviewers’ suggestion, we have expanded the Discussion and included 7 more references in the revised manuscript (please see the marked-up version of the manuscript for comparison).

*2) To further elucidate the specific interaction of IE1-CTD and NCP, the authors introduced the mutation H2A E56R to disrupt binding of IE1-CTD to the acidic patch. The ITC measurements showed no binding of the IE1 CTD to NCPs with this mutation. Nevertheless, binding of LANA to the acidic patch was also significantly disrupted. It would be better if the authors would use/identify a mutation in the NCP that exclusively disrupts the binding of IE1-CTD. Disrupting interaction with ɑ1 of H2B seems to be an obvious mutation target. Did the authors try this? If not we recommend this to be done as the results may strengthen the conclusion that the mode of binding is unique.*

The reviewers’ comment is well taken. We have tried extensively to search for such a mutation but without success. In fact, the initial idea of generating the △476-480 deletion mutant of IE1-CTD was to see the effect of its interaction with α1 of H2B. As it’s shown in Figure 3, the deletion mutant displayed a weakened but still robust interaction with NCP. The next obvious mutation was with Gln44 of H2B, which makes a hydrogen bond with a mainchain carbonyl Thr480. However, Gln44 forms the left bank of zone III (Figure 2) that hosts the binding of both His481 of IE1 and Thr14 of LANA, both of with interact with Gln44 via van der Waals contacts. Hence, mutating this residue is unlikely to affect the interaction of one but not the other with NCP. So far, the H2A E56R mutation is the best we can do, although we haven't completely given up the hope to find such a mutation, possibly also by searching for mutations that affect the interaction between LANA and NCP but not that between IE1 and NCP. Nevertheless, this will be a separate, future work.

*3) Fang et al. report the purification of the full-length hCMV IE1 protein and use the full-length protein in their analytical ultracentrifugation experiment (Figure 3). Unfortunately, they do not include the full-length IE1 in their ITC measurements. Inclusion of full-length IE1 in the ITC measurements would facilitate comparison of affinities with their respective effects in 30-nm chromatin fiber modulation. This should be done.*

We have now measured the NCP binding affinity of full-length IE1 by ITC, and result is included as a supplement to Figure 3—figure supplement 1) in the revised manuscript. The full-length protein alone appeared to misbehave at a low salt concentration (50 mM NaCl) at which ITC measurements for IE1-CTD peptide were performed. To overcome this problem, we performed a new set of ITC experiments with samples including the full-length IE1, IE1-CTD and IE1 lacking CTD at 150 mM salt concentration (Figure 3—figure supplement 1). At this salt concentration, the Kd for IE1-CTD is 2.25 mM, compared to 0.42 mM at 50 mM NaCl, and the Kd for full-length IE1 is 11.33 mM, approximately 5-fold lower than that of IE1-CTD measured at the same salt concentration. IE1 lacking CTD showed no detectable bindings. The lowed NCP binding affinity of full-length could be due to the interference of the extra domains of the full-length protein, and its value relative to that of IE1-CTD is on a par with its behavior in AUC with chromatin fiber.

*4) Several issues about the analytical ultracentrifugation experiments have to be addressed. The authors do not provide an explanation of what the observed difference (between 51.5 S and 48 S) means. Isn't 48 S representative of a condensed chromatin as well? 12_177_601 array in presence of H1 sedimented at 46.6 S in their 2014 Science paper. Where does the discrepancy come from? The statement in paragraph four, Results and Discussion, is not accurate. The mutants only have some moderate 'effect'. At what concentration was this experiment done? Above or below the Kd? As it stands, the effects of the peptide on higher order structure may be over-interpreted, or not significant at all. Further, the authors have to include careful quality control for their 12mer array to check for saturation* – *e.g., by EcoRV digestion. This is especially important for arrays with H1. These issues with the AUC experiments must be clarified before publication can be considered.*

For the issues concerning the sedimentation coefficient, a change of the S-value from 51.5 to 48 by the addition of IE1-CTD indicates that either the molecular weight or the transverse dimension of the sample becomes smaller. It cannot be the first scenario because the binding of IE1-CTD, no matter how small the molecular weight of this peptide, will increase the molecular weight of the sample. And the concentration of IE1-CTD used was 1.2 μM, at a peptide/protein to NCP molar ratio of 5, and approximately 3 times above the Kd value (0.4 μM measured at 50 mM NaCl concentration), and AUC experiments were performed using a buffer without salt, which should make the binding of IE1-CTD to chromatin even stronger. A simple and natural interpretation of the data is the second scenario, in which case the transverse dimension of the chromatin fiber becomes smaller, which is inevitably accompanied by the extension of the longitudinal dimension of the chromatin fiber. Hence, the decrease of the S-value is indicative of a “decondensed” chromatin fiber. Obviously, an S-value of 48 does not correspond to the fully stretched 10-nm nucleosomal array, nevertheless, it is a significant deviation from the fully compacted 30-nm fiber, possibly representing an intermediate state between the two. At present, the molecular details of the intermediate state remain unknown, and we are pursuing single-molecular studies to find out more clues. However, this will be a separate body of work, and we hope to be able to report progress in the near future.

For AUC measurements of 30-nm chromatin fiber, the S values may vary to some extent in different sets of experiments, depending on factors such as the length of DNA repeats, saturation level of the nucleosomal array, batches of purified H1 and batches of nucleosomal array reconstituted. It is our belief that the difference of S-values in the same set of experiments using same batch of reagents is more meaningful than the specific values. Our reconstituted nucleosomal arrays and chromatin fibers are all carefully quality-controlled by both MNase digestion and EM analysis. As shown in the newly added Figure 4—figure supplement 1, the nucleosomal arrays used for reconstituting the 30-nm fiber have nearly identical nucleosome density with > 90% saturation, and they exhibit an extended beads-on-a-string conformation prior to the addition of H1.

In summary, we have added more details about the AUC experiments in the Materials & methods section, included an explanation of the S-value changes in the main text, and provided the results of MNase digestion and EM analysis of the reconstituted nucleosomal arrays according to the reviewers’ suggestions in the revised manuscript.